# Predictors of Survival, Treatment Modalities, and Clinical Outcomes of Diffuse Large B-Cell Lymphoma in Patients Older Than 70 Years Still an Unmet Medical Need in 2024 Based on Real-World Evidence

**DOI:** 10.3390/cancers16081459

**Published:** 2024-04-11

**Authors:** Luís Alberto de Pádua Covas Lage, Rita Novello De Vita, Lucas Bassolli de Oliveira Alves, Mayara D’Auria Jacomassi, Hebert Fabrício Culler, Cadiele Oliana Reichert, Fábio Alessandro de Freitas, Vanderson Rocha, Sheila Aparecida Coelho Siqueira, Renata de Oliveira Costa, Juliana Pereira

**Affiliations:** 1Department of Hematology, Hemotherapy & Cell Therapy, Faculty of Medicine, University of São Paulo (FM-USP), São Paulo 05403000, Braziljuliana.pereira@hc.fm.usp.br (J.P.); 2Laboratory of Medical Investigation in Pathogenesis and Directed Therapy in Onco-Immuno-Hematology (LIM-31), Faculty of Medicine, University of São Paulo (FM-USP), São Paulo 05403000, Brazilfabio.alessandro@alumni.usp.br (F.A.d.F.); 3Fundação Pró-Sangue, Blood Bank of São Paulo, São Paulo 05403000, Brazil; 4Department of Hematology & Hemotherapy, Churchill Hospital, Oxford University, Oxford OX3 7LE, UK; 5Department of Pathology, Faculty of Medicine, University of São Paulo (FM-USP), São Paulo 05403000, Brazil; 6Department of Hematology & Hemotherapy, Faculty of Medicine, Centro Universitário Lusíada (Unilus), Santos 11045101, Brazil; renatadeoliveiracosta@uol.com.br; 7Department of Hematology & Oncology, Hospital Alemão Oswaldo Cruz (HAOC), São Paulo 05403000, Brazil

**Keywords:** diffuse large B-cell lymphoma (DLBCL), old and very old, prognostic factors, clinical outcomes, up-front immunochemotherapy, less intensified regimens

## Abstract

**Simple Summary:**

Here, we assessed clinical outcomes, predictors of survival, and compared responses and toxicities among different therapeutic modalities in a large cohort of older adults with DLBCL. A total of 185 Brazilian patients were included from 2009 to 2020. After a long follow-up, we demonstrated a 5-year OS of 50.2%. The R-MiniCHOP regimen was associated with a lower ORR; however, these patients were more malnourished and had a higher risk on prognostic indexes. Although associated with higher rates of severe neutropenia, the R-CHOP regimen was associated with substantial OS and PFS advantages. We also identified independent prognostic factors associated with poor outcomes, including age ≥ 75 years, high LDH, B-symptoms, clinical stage III/IV, neutrophilia, and reduced lymphocyte/monocyte ratio. Although a significant portion of older DLBCL patients are highly fragile and ineligible for enhanced regimens, attenuated protocols promoted remarkably inferior outcomes compared to those achieved by the R-CHOP.

**Abstract:**

Background: Diffuse large B-cell lymphoma (DLBCL) especially affects the older population. Old (≥60 years) and very old age (≥80 years) DLBCL patients often present high-risk molecular alterations, lower tolerability to conventional immunochemotherapy, and poor clinical outcomes. In this scenario, attenuated therapeutic strategies, such as the R-MiniCHOP and R-MiniCHOP of the elderly regimens, have emerged for this particularly fragile population. However, the responses, clinical outcomes, and toxicities of these regimens currently remain poorly understood, mainly because these individuals are not usually included in controlled clinical trials. Methods: This retrospective, observational, and single-center real-world study included 185 DLBCL, NOS patients older than 70 years treated at the largest oncology center in Latin America from 2009 to 2020. We aimed to assess the outcomes, determine survival predictors, and compare responses and toxicities between three different primary therapeutic strategies, including the conventional R-CHOP regimen and the attenuated R-MiniCHOP and R-MiniCHOP of the elderly protocols. Results: The median age at diagnosis was 75 years (70–97 years), and 58.9% were female. Comorbidities were prevalent, including 19.5% with immobility, 28.1% with malnutrition, and 24.8% with polypharmacy. Advanced clinical stage was observed in 72.4%, 48.6% had bulky disease ≥7 cm, 63.2% had B-symptoms, and 67.0% presented intermediate–high/high-risk IPI. With a median follow-up of 6.3 years, the estimated 5-year OS and PFS were 50.2% and 44.6%, respectively. The R-MiniCHOP of the elderly regimen had a lower ORR (*p* = 0.040); however, patients in this group had higher rates of unfavorable clinical and laboratory findings, including hypoalbuminemia (*p* = 0.001), IPI ≥ 3 (*p* = 0.013), and NCCN-IPI ≥ 3 (*p* = 0.002). Although associated with higher rates of severe neutropenia (*p* = 0.003), the R-CHOP regimen promoted increased OS (*p* = 0.003) and PFS (*p* = 0.005) in comparison to the attenuated protocols. Additionally, age ≥ 75 years, high levels of LDH, B-symptoms, advanced clinical stage (III/IV), neutrophilia, and low lymphocyte/monocyte ratio were identified as poor prognostic factors in this cohort. Conclusions: In this large and real-life Latin American cohort, we demonstrated that patients with DLBCL, NOS older than 70 years still do not have satisfactory clinical outcomes in 2024, with half of cases not reaching 5 years of life expectancy after diagnosis. Although the conventional R-CHOP offers response and survival advantages over attenuated regimens, its myelotoxicity is not negligible. Therefore, the outcomes reported and the prognostic factors here identified may assist clinicians in the appropriate selection of therapeutic strategies adapted to the risk for old and very old DLBCL patients.

## 1. Introduction

Diffuse large B-cell lymphoma (DLBCL) is the most common subtype of non-Hodgkin’s lymphoma (NHL), accounting for 30–40% of all lymphoid malignancies. It is characterized by the monoclonal proliferation of large and pleomorphic lymphoid B-cells, with a diffuse histopathological growth pattern, and origin in the germinal center (centroblasts) or post-germinal center (immunoblasts) [1,2]. DLBCL has wide clinical and biological diversity, and also a remarkably heterogeneous prognosis [2]. Although it is aggressive in nature, with an expected survival of less than 1 year without treatment, DLBCL is potentially curable with immunochemotherapy based on the R-CHOP (rituximab, cyclophosphamide, hydroxydoxorubicin, vincristine, and prednisone) regimen [3,4]. The R-CHOP regimen has high efficacy; however, up to 30–40% of patients will not respond or will relapse, particularly individuals older than 60 years and with other adverse prognostic factors [3,4].

DLBCL especially affects older adults, with an increasing incidence with advancing age [5,6]. Aging is the main risk factor for the development of cancer, and the prognosis of hematological malignancies is notably worse in older adults with cancer when compared to young individuals [7,8,9]. DLBCL of old (≥60 years) and very old (≥80 years) patients has numerous particularities, including high-risk genetic–molecular findings, such as a higher frequency of complex karyotype and higher tumor mutational burden, greater association with infection by Epstein–Barr virus (EBV), lower tolerability to conventional immunochemotherapy, higher expression of genes associated with multidrug resistance (MDR), and, consequently, worse clinical outcomes [10].

With a median age above 70 years at DLBCL diagnosis, older adults with cancer usually have comorbidities that preclude the administration of full doses of standard chemotherapy, as well as having altered pharmacodynamics, a greater tendency to drug toxicity, and an increased probability of mortality associated with treatment [11,12]. Older individuals react less effectively to adverse events resulting from polychemotherapy administration, presenting lower therapeutic tolerability. Physiological changes associated with aging predispose to mucositis, higher rates of infectious complications, greater cardiotoxicity, acute renal injury due to reduced clearance of antineoplastic agents, and greater occurrence of anemia, neutropenia, and thrombocytopenia, resulting from the myelosuppressive effect of cytostatic drugs [13,14].

Therefore, when choosing treatment for old and very old age adults with DLBCL, we must consider several factors, including the individual’s life expectancy, the chemo sensitivity of their neoplasm, pre-existing comorbidities, and the presence of geriatric syndromes, such as instability, immobility, polypharmacy, neurological impairment, and frailty. To this end, several geriatric scales have been developed, such as the Comprehensive Geriatric Assessment (CGA), which considers the functional, nutritional status, mobility, sensory conditions, cognition and mood, social support, and the environment in which the patient is inserted. However, the application of these scales is sometimes complex and difficult, dependent on a qualified and trained professional, requiring extensive time for its use, constituting little-used and impractical tools [15].

Currently, standard-of-care therapy for DLBCL patients of all ages with early-stage disease (I/II) is based on four to six cycles of R-CHOP-21 followed or not by involved-field radiotherapy (IF-RT) with 30-36 Gy. For those individuals with advanced-stage disease (III/IV), six to eight cycles of R-CHOP-21 are applied; however, for patients with advanced disease and International Prognostic Index (IPI) ≥ 3 (intermediate–high or high-risk), Pola-R-CHP (polatuzumab vedotina, rituximab, cyclophosphamide, hydroxydoxorubicin, and prednisone) has become the current new standard of care [3,16,17]. For octogenarians, old individuals with serious comorbidities, or very frail patients, suggested attenuated and/or non-anthracycline-containing immunochemotherapy regimens include the R-MiniCHOP (R-CHOP with 50% dose reduction of doxorubicin [25 mg/sqm I.V. on D1]), R-MiniCHOP of the elderly (rituximab 375 mg/sqm I.V. on D1, cyclophosphamide 400 mg/sqm I.V. on D1, hydroxydoxorubicin 25 mg/sqm I.V. on D1, vincristine 1 mg [fixed dose] I.V. on D1, and prednisone 40 mg/sqm P.O. on days 1 to 5), R-GEMOX (rituximab, gemcitabine, and oxaliplatin), R-CEOP (rituximab, cyclophosphamide, etoposide, vincristine, and prednisone), R-COMP (rituximab, cyclophosphamide, non-pegylated liposomal doxorubicin, vinblastine, and prednisone), R-GCVP (rituximab, gemcitabine, cyclophosphamide, vincristine, and prednisone), R-CEPP (rituximab, cyclophosphamide, etoposide, procarbazine, and prednisone), and BR (bendamustine plus rituximab) [12,18,19]. Although potentially curable, such attenuated immunochemotherapeutic regimens offer decreased responses and poorer outcomes when compared to the conventional R-CHOP-21. However, we emphasize that older individuals are usually underrepresented in clinical trials and that the balance between efficacy and tolerability of alternative regimens to full-dose R-CHOP is also not well established in the current literature.

The World Health Organization (WHO) estimates 80% of older adults by 2050, and population aging is not only a global demographic trend but is particularly prominent in low- and middle-income countries due to the accelerated demographic shift characterized by the inversion of the age pyramids observed in these locations over the last three decades. Based on this premise, the current study aims to describe clinical and laboratory characteristics, assess outcomes, determine predictors of survival, and compare responses and toxicities between different up-front immunochemotherapeutic regimens based on anthracyclines (conventional [R-CHOP-21] and attenuated [R-MiniCHOP and R-MiniCHOP] of the elderly) in a large cohort of older adults than 70 years with DLBCL, NOS.

## 2. Material and Methods

### 2.1. Study Design and Ethical Issues

This is an observational, retrospective, and single-center study carried out at the Instituto do Câncer do Estado de São Paulo (ICESP)/Hospital das Clínicas, Faculty of Medicine, University of São Paulo (HC-FMUSP), Brazil. It was approved by the local Ethics Committee in 2021 (CAAE number: 48897221.2.0000.0068), with the application of the Free and Informed Consent Form (FICF) to all living participants included in the study and with their respective consents. The waiver of the FICF application was obtained for dead patients and with non-recoverable contact. All clinical–demographic, laboratory, pathological, imaging, and therapeutic data were obtained from institutional electronic medical records.

### 2.2. Patients, Elegibility Criteria, and Staging Procedures

Considering all 2798 patients diagnosed with NHL registered in the ICESP/HC-FMUSP Non-Hodgkin’s Lymphoma Database, 1495 patients diagnosed with diffuse large B-cell lymphoma, not otherwise specified (DLBCL, NOS) were initially selected. Of these, 205 (13.7%) were aged ≥ 70 years. Finally, 185/205 (90.2%) were included in this study because they presented clinical–demographic and evolutionary data considered satisfactory for survival analysis. Eligibility criteria included age ≥ 70 years, patients diagnosed and treated at ICESP/HC-FMUSP from January 2009 to December 2020, and biopsy-proven diagnosis of DLBCL, NOS according to the criteria proposed by the World Health Organization Classification for Lymphoid and Hematopoietic Tissue Neoplasms published in 2016 (WHO-2016) [1]. Patients diagnosed with DLBCL from immunoprivileged sites (primary central nervous system lymphoma [PCNSL] or primary testicular DLBCL), high-grade B-cell lymphoma associated with immunosuppression (human acquired immunodeficiency syndrome [AIDS] or post-transplant lymphoproliferative disorders [PTLD]), primary mediastinal DLBCL (PML), T-cell/histiocyte-rich DLBCL (T/H DLBCL), leg-type DLBCL, and low-grade non-Hodgkin lymphomas with histological transformation to DLBCL were excluded from this analysis.

Clinical–epidemiological, laboratory, and histopathological data accessed at diagnosis included age, gender, ethnicity, Eastern Cooperative Oncology Group (ECOG) performance status, Ann Arbor/Cotswold clinical stage, bone marrow (BM) infiltration, central nervous system (CNS) involvement, bulky disease (≥ 7 cm), B-symptoms, immobility, polypharmacy (regular use of ≥ 5 medications), elderly body mass index (E-BMI), and comorbidity profile. Charlson Comorbidity Index (CCI) score was calculated for all patients according to the literature [20,21]. The International Prognostic Index (IPI), Revised International Prognostic Index (R-IPI), and National Comprehensive Cancer Network International Prognostic Index (NCCN-IPI) scores were calculated for all cases as originally described [22,23,24]. Socio-demographic data, such as educational level and family support, were also obtained from the patients’ electronic medical records by a specific researcher. Laboratory variables related to blood cell count (hemoglobin, WBC, neutrophils, lymphocytes, monocytes, neutrophil/lymphocyte ratio [N/Ly], lymphocyte/monocyte ratio [Ly/Mo], and total platelet count), lactic dehydrogenase (LDH), β2-microglobulin, albumin, globulins, and viral serological status for HIV, hepatitis B, and C were also accessed. The lymphoma’s cell of origin (COO) was accessed using the Hans immunohistochemical algorithm [25] in all cases with available formalin-fixed paraffin-embedded (FFPE) material.

Date of diagnosis, complete response (CR), disease relapse and progression, first and last cycle of chemotherapy and radiotherapy, date and cause of death, and date of last follow-up were obtained to assess the overall response rate (ORR), complete response (CR), partial response (PR), overall survival (OS), progression-free survival (PFS), and disease-free survival (DFS). At diagnosis, all DLBCL, NOS patients underwent a complete blood count, comprehensive biochemical tests, including renal and hepatic function, and tests for tumor lysis syndrome (TLS), as well as LDH and β2-microglobulin measurements, and serology for HIV, hepatitis B, and C. All patients eligible for anthracycline-based induction regimens underwent transthoracic echocardiography to estimate left ventricular function (LVF).

Staging was carried out with contrast-enhanced computed tomography (CT) scans of the neck, chest, abdomen, and pelvis, or preferably by CT with positron emission with 18-fluorodeoxyglucose (18-FDG-PET-CT). Unilateral BM biopsy with immunohistochemistry (IHC) was performed in all cases not staged with 18-FDG-PET-CT. Patients with gastrointestinal symptoms underwent upper gastrointestinal endoscopy (UGE) and/or colonoscopy with biopsies. Patients with involvement of ≥ 2 extranodal sites and intermediate–high or high-risk IPI, as well as those with paranasal sinus involvement, paravertebral mass with foraminal invasion, involvement of kidneys, adrenal glands, testicles, ovaries, or breasts had cerebrospinal fluid (CSF) puncture with chemo cytological analysis as part of the staging procedures to exclude secondary CNS involvement.

### 2.3. Up-front Therapy, Response Assessment, and Follow-Up

Up-front treatment of newly diagnosed DLBCL, NOS patients older than 70 years included in this study followed the current institutional protocols at the time of diagnosis. The R-CHOP-21 regimen (rituximab 375 mg/sqm I.V. on D1, cyclophosphamide 750 mg/sqm I.V. on D1, hydroxydoxorubicin 50 mg/sqm I.V. on D1, vincristine 1.4 mg/sqm [maximum 2.0 mg] I.V. on D1, and prednisone 100 mg/day P.O. on days 1 to 5) was administered for 4 to 8 cycles with an interval of 21 days to patients aged between 70 and 80 years, ECOG < 2, Karnofsky Performance Status (KPS) > 70, low index of comorbidities, and without severe organic dysfunction (“fit” patients). Variants of this regimen included R-MiniCHOP or modified R-CHOP-21 (standard R-CHOP-21 with reduced doxorubicin dose to 25 mg/sqm I.V. on D1) and R-mini-CHOP-21 of the elderly (rituximab 375 mg/sqm I.V. on D1, cyclophosphamide 400 mg/sqm I.V. on D1, doxorubicin 25 mg/sqm I.V. on D1, vincristine 1 mg fixed dose I.V. on D1, and prednisone 40 mg/sqm P.O. on days 1 to 5). The R-MiniCHOP regimen was administered for 6 to 8 cycles for patients aged between 70 and 80 years, ECOG 2–3, KPS 50–70, or ≥2 relevant clinical comorbidities (“unfit” patients). The R-mini-CHOP-21 of the elderly regimen was administered for 6 cycles for patients aged ≥ 80 years or for those aged between 70 and 80 years, ECOG 3–4, KPS < 50, and/or ≥2 serious clinical comorbidities (“frail” cases). A minority of frail patients were alternatively treated with non-anthracycline-containing regimens, such as the R-CVP regimen (rituximab 375 mg/sqm I.V. on D1, cyclophosphamide 750 mg/sqm I.V. on D1, vincristine 1.4 mg/sqm [maximum 2.0 mg] I.V. on D1, and prednisone 100 mg/day P.O. on days 1 to 5) for 6 to 8 cycles with non-curative intent. The latter was used in patients with absolute contraindications to anthracyclic agents (symptomatic heart failure or left ventricular ejection fraction < 30%), as well as for very frail individuals (ECOG 4, KPS < 50, and with severe organic dysfunctions).

A pre-phase with CVP (cyclophosphamide 300 mg/sqm I.V. on D1, vincristine 1 mg fixed dose I.V. on D1, and prednisone 40 mg/sqm P.O. on days 1 to 7) was used in frail patients, with ECOG > 2 at diagnosis, KPS < 70, involvement of the gastrointestinal tract with risk of perforation or with high tumor burden (bulky ≥ 7 cm), and risk of tumor lysis syndrome (TLS). This cytoreductive regimen was used for 1 cycle given 7–14 days before curative intent immunochemotherapy. Patients with paranasal sinus involvement, paravertebral lesion with foraminal invasion, involvement of kidneys, adrenal glands, testicles, ovaries, breasts, or with involvement of ≥2 extranodal sites and intermediate–high or high-risk IPI received CNS prophylaxis with intrathecal infusion of 12 mg methotrexate and 2 mg dexamethasone during the first 4 cycles of R-CHOP-like chemotherapy. Involved-field radiotherapy (IF-RT) with 30–36 Gy was administered as consolidation therapy for bulky disease (≥7 cm), early-stage disease, or single mass with residual 18-FDG uptake on PET-CT at the end of immunochemotherapy.

Prophylaxis for infection was unified across the three main different treatment regimens and included use of granulocyte growth factors (G-CSFs) for individuals with absolute neutropenia < 1.0 × 10^9^/L, as well as universal prophylaxis for pneumocystosis with trimethoprim–sulfamethoxazole 400/80 mg P.O. per day and secondary prophylaxis for herpes zoster with acyclovir 400 mg P.O. twice per day. Antibacterial prophylaxis with quinolones was not routinely adopted for either treatment group.

Response assessment was performed after the fourth cycle for interim evaluation and after the last cycle of immunochemotherapy or radiotherapy using CT or 18-FDG-PETCT as recommended by Cheson et al [26]. Clinical follow-up was performed every 3 months in the first two years after complete remission, every 4 months in the third year, every 6 months in the fourth year, and annually after the fifth year.

### 2.4. Histopathological Diagnosis

All cases were diagnosed by two experts in hematopathology and categorized as DLBCL, NOS according to the World Health Organization Classification for Hematopoietic and Lymphoid Tissue Neoplasms published in 2016 (WHO-2016) [1]. The tissue biopsies obtained at diagnosis were kept in formalin-fixed paraffin-embedded (FFPE) material. FFPE sections of 5 µm were displayed on silanized slides and stained with hematoxylin and eosin (HE) for initial analysis. Immunohistochemistry (IHC) was carried out with monoclonal antibodies CD45 (Dako, 2B11+PD7/26, 1/2000, Glostruo, Denmark), pan-B CD20 (Dako, L26, 1/1000), pan-T CD3 (Dako, F7.2.38, 1/500), proliferative marker Ki-67 (Dako, J55, 1/1600), CD10 (Novocastra, S6C6, 1/2000, Newcastle, UK, BCL6 (Abcam, EPR11410-43, 1/500, Boston, MA, USA), and MUM-1/IRF-4 (Abcam, EPR5653, 1/500). The immunohistochemical algorithm proposed by Hans et al was used to determine the cell of origin (COO), analyzing the expression pattern of the markers CD10, BCL6, and MUM-1/IRF-4 [25].

### 2.5. Statistical Analysis

Data were shown in accordance with the variables evaluated. Categorical variables were presented in absolute (N) and relative (%) values. Numerical variables were presented as measures of central tendency (median), dispersion (min–max range), and position. The median follow-up time was calculated using the reverse Kaplan–Meier method (reverse KM). Analysis of overall survival (OS), progression-free survival (PFS), and disease-free survival (DFS) was performed using the Kaplan–Meier (KM) method, and the log-rank test was used to establish associations between variables and outcomes. The Chi-square test and the Kruskal–Wallis test were applied to assess statistically significant differences in clinical–demographic characteristics and responses between different treatment modalities. OS was considered from the date of diagnosis to death, PFS from the date of diagnosis to disease progression, death, or last follow-up, and DFS from the date of complete response (CR) to disease progression, death, or last follow-up. Data were censored at the last follow-up.

Analysis to determine predictors for outcomes, including those related with OS and PFS, was performed using Cox’s semiparametric univariate method. Multivariate analysis using a multi-step Cox’s regression model was conducted to determine independent prognostic variables. All variables with a *p*-value ≤ 0.10 identified in univariate analysis were included in the final model for multivariate analysis. The results were presented in hazard ratio (HR) and 95% confidence interval (95% CI). All analyses were performed using the R-statistical software version 4.1.3 for Windows and a *p*-value ≤ 0.05 was assigned as statistically significant.

## 3. Results

### 3.1. Clinical, Epidemiological, Laboratorial, and Histopathological Findings

The clinical–demographic, laboratory, and pathological findings of the 185 patients with DLBCL, NOS are summarized in Table 1. The median age was 75 years (range: 70–97 years). A total of 109 patients (58.9%) were female, 76.7% (142) were Caucasian, and only 7.6% (7/92) had tertiary education. The median Charlson Comorbidity Index (CCI) score was 6 points (range: 5–10 points), and 63.2% (117) had at least one relevant comorbidity. More than a quarter of the whole cohort (25.4%; 47) had diabetes mellitus, 7% (13) had congestive heart failure, and 6.5% (12) had chronic obstructive pulmonary disease (COPD), as well as cerebrovascular disease (6.5%; 12) or secondary malignancy (6.5%; 12). Synchronous thrombosis at the diagnosis of DLBCL was observed in 11.4% (21) of cases. Immobilism occurred in 19.5% (36) of cases, and polypharmacy, defined as regular use of ≥ 5 medications, was observed in 24.8% (46). According to the Karnofsky Performance Status, 56.2% (104) of patients were categorized as “fit” (KPS > 70), 27.6% (51) as “unfit” (KPS 50–70), and 16.2% (30) as “frail” (KPS < 50). The median for body mass index (BMI) was 25.2 (range: 16.1–40.2), with malnutrition (geriatric BMI < 22) occurring in 28.1% (52) and overweight (geriatric BMI > 27) observed in 34.1% (63). Advanced-stage disease (Ann Arbor/Cotswold III/IV) was observed in 72.4% (134), 48.6% (90) had bulky disease ≥ 7 cm, 63.2% (117) had B-symptoms, 76.2% (141) had extranodal involvement, and 37.2% (69) had ≥ 2 extranodal sites involved by NHL. Bone marrow infiltration was demonstrated in only 3.8% (7) of cases, as well as CNS involvement. Patients with intermediate–high and high-risk NCCN IPI corresponded to the majority of our cohort (88.1%;163), as well as patients with intermediate–high and high-risk IPI, corresponding to 67% (124) of cases.

The medians of hemoglobin, total white blood cell (WBC) count, neutrophils, lymphocytes, monocytes, and platelets were 121 g/L, 7.82 × 10^9^/L, 5.19 × 10^9^/L, 1.30 × 10^9^/L, 0.70 × 10^9^/L, and 256 × 10^9^/L, respectively. The medians of the neutrophil/lymphocyte (N/Ly) and lymphocyte/monocyte (Ly/Mo) ratios were 3.0 and 2.0, respectively. The medians of the patient LDH/control LDH ratio, β2-microglobulin, albumin, and total globulins were 1.0, 3.2 mg/dL, 3.9 g/dL, and 2.8 g/dL, respectively. Anemia occurred in 15.2% (28/184) of cases, leukocytosis in 14.1% (26/184), neutrophilia in 20.1% (37/184), lymphocytosis in 6% (11/184), and monocytosis in 16.3% (30/184). More than a quarter of the whole cohort (25.9%; 48) had a patient LDH/control LDH ratio ≥ 1.5, hypoalbuminemia (<3.5 g/dL) was observed in 28.6% (52/182), and hypogammaglobulinemia (<1.5 g/dL) occurred in only 1.1% (2/180) of cases. Additionally, positivity for HBsAg, total anti-HBc, and anti-HCV occurred in 0% (0/176), 15.9% (28/176), and 4.8% (8/171) of cases, respectively. The immunohistochemical algorithm proposed by Hans et al. could be applied in 58.9% (109) of cases [25]. Of these, 52.3% (57) had a germinal center (GCB)-like phenotype and 47.7% (52) had an activated B-cell (ABC)-like phenotype.

### 3.2. Up-Front Therapy for Elderly DLBCL, NOS Patients: Modalities, Responses, and Toxicities

Among the 185 DLBCL older than 70 years patients NOS included in this study, 1.6% (3) did not experience any antineoplastic therapy, because they died before starting first-line treatment. Among the 182 (98.4%) effectively treated cases, 57.1% (104) received the R-CHOP-21 regimen as induction, 18.0% (51) received the R-Mini-CHOP/modified R-CHOP-21 regimen, 13.2% (24) received the R-Mini-CHOP-21 of the elderly regimen, and only 1.7% (3) were treated without curative intent with the R-CVP regimen. Cytoreduction based on the CVP regimen was used in 65.9% (120) of treated cases, while CNS prophylaxis with the administration of intrathecal chemotherapy was applied in 50.5% (92) of cases. Seventy-six patients (41.8%) experienced IF-RT as an adjunct to primary therapy.

The overall response rate (ORR) for the whole cohort was 68.1% (95% CI: 60.8–74.8%), with complete response (CR) achieved in 65.9% (95% CI: 58.5–72.7%) and partial response (PR) in 2.2% (95% CI: 0.6–5.5%). Primary refractory disease occurred in 13.2% (95% CI: 8.6–18.9%) of cases, including 1.1% (2) with stable disease (SD) and 12.1% (22) with persistent or progressive disease (PD) at the end of up-front immunochemotherapy. The mortality rate (MR) during first-line therapy was 18.7% (95% CI: 13.3–25.1%). Among these 34 early deaths, 70.6% (24) occurred due to infectious complications, 17.6% (6) were due to disease progression, and in 11.8% (4), the cause of death could not be defined, usually in cases with death registered outside the oncological service of origin. The overall mortality rate (OMR) during all follow-up was 52.4% (97), with disease progression as the main cause of death (56; 57.7%). The early mortality rate (EMR), characterized as the percentage of deaths during the first 100 days after the DLBCL diagnosis, was 9.7% (18), occurring mainly due to infectious complications (10; 55.6%).

Table 2 summarizes the main up-front treatment modalities, clinical–demographic characteristics, responses, and toxicities of each treatment arm. The R-mini-CHOP of the elderly regimen had a lower overall response rate (ORR) (*p* = 0.040). However, as expected, patients in this group were not only older, but had other unfavorable clinical–laboratory characteristics, including higher rates of hypoalbuminemia (*p* = 0.001), and higher frequency of cases with IPI ≥ 3 (*p* = 0.013) and NCCN-IPI ≥ 3 (*p* = 0.002). The CCI scores were well balanced between the treatment groups. No differences were observed regarding thrombocytopenia, febrile neutropenia, therapy interruption, and mortality during first-line therapy, although grade 3/4 neutropenia was more common in patients treated with R-CHOP-21 (*p* = 0.003).

### 3.3. Clinical Outcomes

The median follow-up for the whole cohort was 6.3 years (95% CI: 5.5–6.7 years). The median overall survival (OS) was 5.3 years (95% CI: 2.5–8.8 years), with an estimated 2-year OS of 61.6% (95% CI: 54.1–68.2%) and 50.2% (95% CI: 42.4–57.5%) in 5 years (Figure 1A). The median progression-free survival (PFS) was 3.7 years (95% CI: 2.0–5.8 years), and the estimated 2-year and 5-year PFS were 57.5% (95% CI: 50.0–64.3%) and 44.6% (95% CI: 37.0–51.9%), respectively (Figure 1B). Additionally, the median disease-free survival (DFS) was 9.8 years (95% CI: 5.8—not reached), with an estimated 2-year and 5-year DFS of 79.7% (95% CI: 70.2–84.6%) and 63.4% (95% CI: 52.0–68.2%), respectively (Figure 1C).

Considering the age at diagnosis, 69.7% (129) of cases were diagnosed between 70 and 80 years and 30.3% (56) over 80 years. The median OS was 5.5 years (95% CI: 4.5–6.6 years) for patients aged between 70 and 80 years, and 3.6 years (95% CI: 2.3–4.9 years) for patients over 80 years (*p* = 0.05). The estimated 2-year and 5-year OS were 77.9% (95% CI: 69.2–87.8%) and 64.8% (95% CI: 54.0–77.7%), respectively, for the group aged between 70 and 80 years, and 43.8% (95% CI: 31.0–62.0%) and 32.0% (95% CI: 20.0–51.0%), respectively, for the group aged over 80 years, *p* = 0.05.

Concerning the clinical outcomes stratified by up-front therapeutic modality, the median OS for patients treated with R-CHOP, R-MiniCHOP, and R-MiniCHOP of the elderly were 8.9 years (95% CI: 7.7–10.1 years), 6.0 years (95% CI: 4.4–7.6 years), and 2.9 years (95% CI: 1.2–4.7 years), respectively, *p* = 0.003. The estimated 2-year and 5-year OS were 82.0% (95% CI: 73.9–91.2%) and 71.6% (95% CI: 61.0–84.0%) for the R-CHOP group, 67.5% (95% CI: 51.5–88.4%) and 57.8% (95% CI: 41.0–81.6%) for the R-MiniCHOP group, and 48.1% (95% CI: 24.5–94.1%) and 24.1% (95% CI: 5.1–71.3%) for the R-MiniCHOP of the elderly group, *p* = 0.003 (Figure 2A). The median PFS for patients treated with R-CHOP, R-MiniCHOP, and R-MiniCHOP of the elderly were 5.6 years (95% CI: 4.5–6.7 years), 4.4 years (95% CI: 3.0–5.7 years), and 1.7 years (95% CI: 0.6–2.9 years), respectively, *p* = 0.005. The estimated 2-year and 5-year PFS were 61.1% (95% CI: 51.8–72.2%) and 45.7% (95% CI: 36.2–57.8%) for the R-CHOP group, 56.4% (95% CI: 42.1–75.5%) and 40.7% (95% CI: 26.3–63.1%) for the R-MiniCHOP group, and 20.5% (95% CI: 6.6–63.5%) and 10.2% (95% CI: 1.7–61.3%) for the R-MiniCHOP of the elderly group, *p* = 0.005 [Figure 2B].

### 3.4. Prognostic Factors: Univariate and Multivariate Analysis

In the univariate analysis, the variables associated with decreased OS were age ≥ 75 years [HR: 2.08, 95% CI: 1.35–3.20, *p* = 0.001], non-Caucasian ethnicity [HR: 1.80, 95% CI: 1.14–2.83, *p* = 0.012], neurologic comorbidity [HR: 2.37, 95% CI: 1.19–4.72, *p* = 0.014], LDH > UVN [HR: 1.60, 95% CI: 1.07–2.41, *p* = 0.023], leukocytosis ≥ 11.0 × 10^9^/L [HR: 2.61, 95% CI: 1.59–4.28, *p* < 0.001], neutrophilia ≥ 7.0 × 10^9^/L [HR: 2.32, 95% CI: 1.46–3.67, *p* < 0.001], monocytosis ≥ 1.0 × 10^9^/L [HR: 2.01, 95% CI: 1.17–3.02, *p* < 0.001], Ly/Mo ratio < median [HR: 2.19, 95% CI: 1.32–3.64, *p* = 0.002], hypoalbuminemia [HR: 1.59, 95% CI: 1.03–2.46, *p* = 0.036], advanced-stage III/IV [HR: 2.36, 95% CI: 1.38–4.04, *p* = 0.002], B-symptoms [HR: 1.89, 95% CI: 1.21–2.93, *p* = 0.005], and involvement of ≥ 2 extranodal sites [HR: 2.73, 95% CI: 1.19–6.26, *p* = 0.018]. 

Predictors for decreased PFS were age ≥ 75 years [HR: 2.00, 95% CI: 1.33–3.01, *p* = 0.001], non-Caucasian ethnicity [HR: 1.68, 95% CI: 1.09–2.60, *p* = 0.020], neurologic comorbidity [HR: 2.76, 95% CI: 1.38–5.49, *p* = 0.004], LDH > UVN [HR: 1.48, 95% CI: 1.00–2.17, *p* = 0.048], leukocytosis ≥ 11.0 × 10^9^/L [HR: 2.26, 95% CI: 1.38–3.69, *p* = 0.001], neutrophilia ≥ 7.0 × 10^9^/L [HR: 2.09, 95% CI: 1.34–3.27, *p* = 0.001], monocytosis ≥ 1.0 × 10^9^/L [HR: 1.97, 95% CI: 1.24–2.73, *p* = 0.007], Ly/Mo ratio < median [HR: 2.04, 95% CI: 1.25–3.33, *p* = 0.004], hypoalbuminemia [HR: 1.50, 95% CI: 1.08–2.27, *p* = 0.049], advanced-stage III/IV [HR: 2.58, 95% CI: 1.53–4.34, *p* < 0.001], B-symptoms [HR: 1.84, 95% CI: 1.21–2.80, *p* = 0.004], and involvement of ≥2 extranodal sites by NHL [HR: 2.38, 95% CI: 1.04–5.45, *p* = 0.039].

In the multivariate analysis, the independent prognostic factors associated with decreased OS in our older DLBCL, NOS cohort were age ≥ 75 years [HR: 2.22, 95% CI: 1.42–3.47, *p* = 0.001], neutrophilia ≥ 7.0 × 10^9^/L [HR: 2.18, 95% CI: 1.23–3.84, *p* = 0.007], Ly/Mo ratio < median [HR: 1.98, 95% CI: 1.18–3.32, *p* = 0.010], and advanced Ann Arbor clinical stage (III/IV) [HR: 2.36, 95% CI: 1.34–4.13, *p* = 0.003]. Similarly, age ≥ 75 years [HR: 2.33, 95% CI: 1.52–3.58, *p* < 0.001], high LDH (>UVN) [HR: 1.09, 95% CI: 1.00–1.18, *p* = 0.047], B-symptoms [HR: 1.62, 95% CI: 1.04–2.52, *p* = 0.032], advanced-stage III/IV [HR: 2.38, 95% CI: 1.38–4.11, *p* = 0.002], neutrophilia ≥ 7.0 × 10^9^/L [HR: 2.25, 95% CI: 1.41–3.59, *p* = 0.001], and low Ly/Mo ratio [HR: 2.08, 95% CI: 1.25–3.46, *p* = 0.005] were independently associated with decreased PFS. Table 3 summarizes the main prognostic factors for OS and PFS identified in our cohort.

## 4. Discussion

In this study, we reported the main clinical–demographic, laboratory characteristics, clinical outcomes, and survival predictors in a large retrospective cohort of old and very old adults aged over 70 years and with a biopsy-proven diagnosis of DLBCL, NOS treated in a real-life setting in Brazil. We also assessed the response rates and long-term clinical outcomes in this population submitted to different up-front immunochemotherapy strategies based on anthracyclines, aiming to compare the efficacy and toxicity profile between conventional therapy with full-dose R-CHOP-21 and attenuated regimens, such as the R-MiniCHOP and R-MiniCHOP of the elderly. The identification of such prognostic factors and the reporting of these outcomes is essential for a better understanding of the DLBCL behavior in older adults in different geographic areas. Our data reflect the characteristics of one of the largest cohorts of older individuals with DLBCL in Latin America. Currently, the clinical–epidemiological characterization and outcomes of cohorts composed of older adults with DLBCL from North America and Europe are well established in the literature; however, data from Latin America, and particularly South America, are virtually unknown. Additionally, the determination of prognostic factors, response patterns, and toxicity profile reported upon treatment with different up-front immunochemotherapeutic regimens may assist in establishing risk-adapted therapeutic strategies for older adults with DLBCL, NOS.

Although the initial management of DLBCL in older adults (aged ≥ 60 years) still remains centered on the R-CHOP-21, an appreciable portion of cases will not be candidates for this regimen due to very old age, poor performance status, or due to severe and limiting comorbidities [3,4,18,27]. Therefore, the primary treatment of DLBCL in old and very old adults is challenging, and to determine the better up-front therapeutic modality, we must weigh the biological aggressiveness of the neoplasm, its high curative potential, and the geriatric vulnerability presented by this population [27]. Given all these particularities, attenuated or low-dose therapeutic strategies based and not based on anthracyclic agents have emerged for the management of unfit or frail older adults with DLBCL. Some of them, particularly non-containing anthracycline regimens, are merely palliative; however, many attenuated regimens are applied with curative intent [12,28]. Two examples of these curative attenuated regimens involve the R-MiniCHOP (or modified R-CHOP, with 50% dose reduction of doxorubicin) and the R-MiniCHOP of the elderly (with approximately 50% dose reduction of all conventional cytotoxic agents) [29]. However, the real impact of these low-dose strategies on the long-term outcomes of older adults with DLBCL is poorly known, mainly because this population is often excluded from prospective and controlled clinical trials. Given this gap in the medical literature, we aimed in this study to address this issue in a real-world scenario.

In our study, the R-MiniCHOP of the elderly regimen showed a significantly decreased overall response rate (ORR) (45.6%) compared to the full-dose R-CHOP (72.1%) and R-MiniCHOP (70.6%), *p* = 0.040. However, mortality during up-front therapy was not different between patients treated with the three regimens (17.3% for R-CHOP, 19.6% for R-MiniCHOP, and 25.0% for R-MiniCHOP of the elderly, *p* = 0.681). It is worth mentioning that individuals treated with the R-MiniCHOP of the elderly regimen, in addition to being older (median ages of 73 years for R-CHOP, 78 years for R-MiniCHOP, and 83 years for R-MiniCHOP of the elderly), had numerous unfavorable clinical–biological characteristics, including a higher percentage of malnutrition (*p* = 0.001), and a higher percentage of individuals with IPI (*p* = 0.013) and NCCN-IPI (*p* = 0.002) categorized as intermediate–high and high risk, as demonstrated in Table 2. Certainly, this poor clinical–biological characterization contributed to decreased ORR observed in the arm treated with the R-MiniCHOP of the elderly. Interestingly, severe hematological toxicities, including the occurrence of grade 3/4 thrombocytopenia (*p* = 0.238) and febrile neutropenia (*p* = 0.907), did not differ between the three treatment arms. Despite a higher occurrence of grade 3/4 neutropenia in individuals treated with full-dose R-CHOP (*p* = 0.003), this did not translate into higher rates of febrile neutropenia (*p* = 0.907), treatment interruption (*p* = 0.671), or mortality during the induction (*p* = 0.681). We believe that the universal prophylactic use of white cell growth factors adopted in our cohort based on neutrophil counts < 1.0 × 10^9^/L may have strongly contributed to these favorable results.

Concerning the long-term clinical outcomes, we observed that DLBCL older adults treated with full-dose R-CHOP had benefits in OS and PFS when compared to individuals undergoing the attenuated R-MiniCHOP and R-MiniCHOP of the elderly regimens. The estimated 5-year OS was 71.6% for R-CHOP, 57.8% for R-MiniCHOP, and 24.1% for R-MiniCHOP of the elderly, *p* = 0.003. Similarly, the estimated 5-year PFS was 45.7% for R-CHOP, 40.7% for R-MiniCHOP, and only 10.2% for R-MiniCHOP of the elderly, *p* = 0.005. These outcomes obtained in our real-life study are in agreement with those presented by Meguro et al [30]. These authors compared the efficacy of the conventional R-CHOP-21 regimen (n = 69) with the attenuated R-70%CHOP regimen (n = 61) in a retrospective cohort of elderly DLBCL patients from Japan. The R-70%CHOP regimen was applied in older adults aged over 70 years, while full-dose R-CHOP was administered to patients aged between 50 and 69 years. Patients treated with R-70%CHOP had worse 3-year OS (58% vs 68%, *p* < 0.05) and 3-year EFS (45% vs 70%, *p* < 0.05) compared to those treated with full-dose R-CHOP. Furthermore, in this study, patients in the R-70%CHOP arm had similar rates of grade 4 thrombocytopenia and grade 4 anemia compared to those submitted to R-CHOP. However, the full-dose R-CHOP group had a higher frequency of grade 4 leukopenia (76.8% vs 60.7%), although the rates of febrile neutropenia and infection were comparable in both groups [30]. In other words, taking into account the methodological differences, as previously revealed by the Japanese group, we demonstrated in our real-life study that attenuated regimens based on anthracyclic agents were associated with decreased outcomes in DLBCL older adults, and although conventional treatment with R-CHOP was associated with higher rates of severe neutropenia, this adverse event may be easily manageable and was not translated into higher rates of infectious complications or higher mortality.

Due to the increased risk of hematological and cardiac toxicity in older adults treated with full-dose R-CHOP, attenuated varieties of this regimen have been recently evaluated. In this sense, Peyrade et al. prospectively evaluated the efficacy of the R-MiniCHOP of the elderly regimen in a multicenter cohort consisting of 150 DLBCL patients older than 80 years. This regimen was shown to be safe and feasible, with an ORR of 73%, and 2-year PFS and OS estimates of 47% and 59%, respectively [29]. Unlike what was demonstrated by Peyrade et al, in our cohort, the outcomes presented by patients treated with the R-MiniCHOP of the elderly regimen were very poor, with an ORR of 45.6%, and estimated 2-year PFS and OS of only 20.5% and 48.1%, respectively [29]. However, we emphasize that our study was conducted retrospectively in a real-life setting, differently from the study of the Franco-Belgian group, which had a prospective design and included highly selected older patients, where individuals with poor performance status (ECOG ≥ 2) or high rates of comorbidities were excluded. The multivariate analysis of the study conducted by Peyrade et al. demonstrated that OS was strongly affected by serum albumin concentrations below 35 g/L (HR: 3.2, 95% CI: 1.4–7.1, *p* = 0.0053) [29]. In our series, 54.2% (13) of patients treated with R-MiniCHOP of the elderly had hypoalbuminemia, which may, at least partially, justify the worse outcomes found in our cohort.

High-level evidence for the management of older adults with DLBCL considered unfit or frail is unavailable in the current medical literature, considering that most prospective studies published to date are single-arm. Therefore, to date, only one meta-analysis has compared the efficacy and safety of different first-line treatments in this population [31]. This meta-analysis included 1839 patients enrolled in 13 clinical trials and demonstrated a PFS advantage for patients treated with the R2CHOP-21 (lenalidomide plus R-CHOP-21) regimen compared to the R-COMP (R-CHOP with non-pegylated liposomal doxorubicin), R-MiniCEOP (with epirubicin replacing doxorubicin), RCHOP-14, and RCHOP-21. No statistically significant differences in ORR, CR, EFS, and OS were demonstrated between the previously mentioned regimens, although the R-COMP regimen promoted lower rates of toxic cardiac events and severe neutropenia [31].

The standard of care for older adults with DLBCL centered on the R-CHOP-21 regimen was established by the Groupe d’Etudes des Lymphomes de l’Adulte (GELA) LNH-98-5, RICOVER-60, LNH03-6B, and UK NCRI R-CHOP14v21 randomized trials, enrolling newly diagnosed DLBCL patients aged between 60 and 80 years. These trials validated the superiority of the R-CHOP regimen versus CHOP polychemotherapy in older adults, with an absolute increase of 13% in OS with the addition of the anti-CD20 monoclonal antibody to the CHOP backbone and demonstrated equivalence in terms of efficacy of the R-CHOP-21 and R-CHOP-14 regimes. However, these clinical trials did not involve octogenarians and included highly selected patients with ECOG ≤ 1 [32,33,34,35,36,37]. As shown in Table 4, the full-dose R-CHOP regimen promoted ORR between 83 and 91% in these studies, and 5-year PFS and OS estimates of around 54–73% and 58–69%, respectively. In our real-world study, the ORR of older adults treated with full-dose R-CHOP-21 was lower (72.1%); however, the estimated 5-year OS of this subgroup (71.6%) was similar to that observed in previously reported trials. Our median DFS was quite long (9.8 years); therefore, we believe that our high early mortality rate (17.3% in the R-CHOP regimen) resulting from the poor conditions presented by our patients at diagnosis adversely impacted the ORR of older adults treated with R-CHOP-21; however, once these patients achieved remission, their responses were durable and promoted OS comparable to that described in other international cohorts.

Regimens containing alternative anthracyclines, such as non-pegylated liposomal formulations of doxorubicin (R-COMP) and epirubicin (R-MiniCEOP), are promising strategies to minimize cardiotoxicity in older adults with DLBCL without substantially compromising therapeutic efficacy [38,39]. For frail patients or those with contraindications to the use of anthracyclic agents, alternative regimens such as R-CEOP (rituximab, cyclophosphamide, etoposide, and prednisone), R-CVP (rituximab, cyclophosphamide, vincristine, and prednisone), BR (bendamustine plus rituximab), or regimens containing gemcitabine (R-GEMOX [rituximab, gemcitabine, and oxaliplatin] and R-GCVP [rituximab, gemcitabine, cyclophosphamide, vincristine, and prednisone]) have been evaluated. Although they are still considered potentially curable, long-term survival usually does not exceed 50%, being lower than that offered by anthracycline-containing regimens [40,41,42,43]. In our study, we did not assess the role of such attenuated regimens not based on doxorubicin, although they constitute interesting alternatives for a specific niche of elderly people with newly diagnosed DLBCL.

Despite obtaining satisfactory responses and outcomes in the population of older adults with DLBCL considered “fit” (ECOG ≤ 2 and KPS > 70) and treated with the full-dose R-CHOP regimen in our study, with an ORR of 72.1% and estimated 2-year OS and PFS of 82% and 61.1%, respectively, the estimated 5-year OS and PFS for the whole cohort composed of 185 patients older than 70 years were poor, corresponding to 50.2% and 44.6%, respectively. These data reflect that currently, in a real-life context, the up-front treatment of DLBCL in old elderly and very old adults is still an unmet medical need, and therefore clinical outcomes can be improved with the use of new therapeutic approaches focused on target drugs, or through the appropriate selection of cases who are candidates for full-dose regimens. We believe that the low survival expectations of our whole cohort were due to the high proportion of frail patients with severe comorbidities, treated in this context with alternative protocols based on attenuated doses of anthracyclines. Although in our cohort we did not categorize a patient’s fitness according to Comprehensive Geriatric Assessment (CGA) recommendations, many individuals were classified as unfit (27.6%) or frail (16.2%) according to the Karnofsky Performance Status (KPS) scale. In addition, the prevalence of other robust markers of geriatric frailty was very high, such as immobility (19.5%), malnutrition (28.1%), and the presence of more than one serious comorbidity (63.2%), making 40.5% (75) of our population ineligible for the full-dose R-CHOP protocol. Table 5 summarizes the main clinical outcomes of old DLBCL patients stratified according to their geriatric vulnerability obtained in other studies, revealing data similar to those demonstrated by us.

New therapeutic approaches focused on improving outcomes for older adults with DLBCL involve combining new agents with the conventional R-CHOP regimen. However, to date, such strategies have not demonstrated a high capacity to increase CR and OS in this population compared to conventional immunochemotherapy based on anthracyclines. Although the combination of lenalidomide (15 mg on days 1–14) plus R-CHOP was tested in a phase II study involving 49 fit patients with DLBCL aged 60–80 years demonstrating promising results, including a CR rate of 86%, and 2-year OS and PFS estimates of 92% and 80%, respectively [49], other combinations involving new drugs were unable to improve outcomes in the geriatric population. In this sense, the REMARC trial evaluated the role of maintenance therapy with lenalidomide in elderly DLBCL patients who responded to R-CHOP. In this study, the PFS increase promoted by maintenance with lenalidomide was very slight and no benefit was observed in OS when this therapy was compared to the placebo group [50]. Additionally, the ibrutinib plus R-CHOP combination was tested in a phase III study involving patients with non-GCB DLBCL with a median age of 62 years. The study also failed to meet its primary endpoint (EFS increase) in the whole cohort and in the subgroup of patients older than 60 years, and the ibrutinib + R-CHOP regimen was associated with substantially increased rates of severe adverse events, including febrile neutropenia and infectious complications, as well as higher rates of therapy discontinuation when compared to cases treated with R-CHOP [51].

In the last five years, great progress has been made in the management of relapsed and refractory DLBCL through the incorporation of new therapeutic strategies, such as chimeric antigen receptor T (CAR-T) cell therapy and bispecific monoclonal antibodies (BsMoAbs). Offering promising responses, bispecific antibodies and CAR-T therapies have also proven to be applicable alternatives in elderly and fit patients with DLBCL. In this sense, a recent study conducted by Ram R et al. (2021) compared clinical outcomes between 41 R/R DLBCL patients older than 70 years and 41 younger patients undergoing CAR-T therapy. The authors demonstrated no differences in the incidence of grade ≥ 3 toxicities, such as cytokine release syndrome (CRS) (*p* = 0.29), neurotoxicity (*p* = 0.54), and duration of hospitalization (*p* = 0.55), between younger and older adults. Additionally, response rates were similar between both groups (*p* = 0.33), and older adults achieved satisfactory outcomes, with 12-month OS and PFS of 69% and 32%, respectively [52].

Similarly, the use of anti-CD20/CD3 bispecific monoclonal antibodies has been expanded to older adults with DLBCL. The toxicity of these agents is usually lower than that of CAR-T therapy, which makes them very promising in the geriatric population, including frail individuals. Among these, epcoritamab is of particular interest in patients aged ≥ 60 years, given its subcutaneous administration. A recent trial conducted by Thieblemont et al. (2023), involving 157 patients with R/R DLBCL, with a median age of 64 years and treated with epcoritamab, demonstrated an ORR of 63% and a CR of 39%. Although CRS occurred frequently, it was usually mild (≤grade 2), being easily manageable in most cases [53]. Another study explored the use of mosunetuzumab as monotherapy for the up-front treatment of old (>60 years) and very old (>80 years) patients who were frail and ineligible for conventional immunochemotherapy. In this study, involving 54 individuals with a median age of 83 years, ORR was 56%, with CR of 43%, and no occurrence of CRS or neurotoxicity ≥ grade 3 [54]. Therefore, we believe that the independent prognostic factors found in our study, combined with the identification of classic biomarkers associated with frailty, such as age ≥ 80 years, malnutrition (albumin < 3.5 g/dL or geriatric BMI < 22), and KPS < 70, may be valuable tools for selecting older adults with DLBCL for new therapies capable of offering promising outcomes, such as those listed above, as well as for attenuated immunochemotherapeutic regimens, such as R-MiniCHOP, R-MiniCHOP of the elderly, and protocols not based on anthracyclines.

Another way to optimize outcomes for older DLBCL patients is based on the systematic use of tools capable of selecting suitable candidates for intensified or attenuated therapeutic approaches, offering personalized strategies adapted to the frailty profile of each individual. In older adults with cancer, the aim of maintaining quality of life can often exceed the aim of promoting a cure or extension of survival. Therefore, Comprehensive Geriatric Assessments (CGAs) can be used for this purpose. CGAs may include assessments of functional status, comorbid medical conditions, cognition, psychological status, social support, nutritional status, and concomitant medications [55]. However, such assessments are quite complex and require a lot of time and are rarely applied in clinical practice. As a result, judging the frailty of older adults with DLBCL has been subjective and based on parameters such as age, comorbidity index, and performance status. Aiming to establish more objective and easier-to-implement parameters, different collaborative groups have proposed adapted CGAs, such as the FIL tool (Italian Lymphoma Foundation) that incorporates the Chronic Illness Rating Scale-Geriatric (CIRS-G), Instrumental Activities of Daily Living (IADL), and Activities of Daily Living (ADL) to categorize patients as fit, unfit, or frail [56]. The FIL tool can assist in the selection of fit and unfit patients for therapies with curative intent and guide clinicians to discuss palliative approaches in frail patients. However, it lacks strong validation in large prospective cohorts and remains little adopted in clinical practice. In our study, we did not apply any tool based on CGAs to stratify the geriatric vulnerability of our patients; however, we categorized our cases into fit, unfit, and frail according to the KPS scale, and we were able to demonstrate OS estimates clearly distinct between individuals with KPS > 70, between 50 and 70, and <50, as well as employing attenuated therapeutic strategies for cases considered unfit or frail according to this scale.

Due to the increasing incidence of chronic diseases with advancing age, approximately 70% of our cohort had one or more comorbidities. A population study conducted in the Netherlands demonstrated a similar prevalence, where 66% of patients with lymphoma aged over 70 years had comorbidities [57]. Additionally, the same group analyzed the severity of comorbidities and showed that patients with NHL and cardiovascular disease had a 3.3 times higher chance of developing treatment toxicity [58]. Interestingly, in our analysis, the presence of cerebrovascular disease was a predictor of worse OS and PFS. We also observed malnutrition in 28.1% of our cases and low albumin levels adversely affected our cohort, as demonstrated in the univariate analysis. Recently, a geriatric nutritional risk index calculated from body weight and serum albumin was an IPI-independent adverse factor predicting OS in older adults with DLBCL [59]. While immobility was observed in almost 20% of our cohort, walking speed has been shown to adversely correlate with survival. With a cumulative risk for the incidence of deep vein thrombosis (DVT) of around 3% in DLBCL patients [60], our cohort presented a rate of 11.4% for thromboembolic events at the diagnosis of DLBCL. Although it appears high, this may be due to the large number of incidental and asymptomatic thrombotic events detected during staging procedures based on CTs and 18-FDG-PETCT. Classically, the prevalence of non-GCB DLBCL increases continuously with age, reaching 40–50% after the age of 60 years, according to data obtained from North American and European cohorts [61,62]. This DLBCL subtype is usually characterized by extranodal involvement, molecular findings associated with immune evasion, the presence of recurrent MYD88 L265P and CD79B mutations, and worse clinical outcomes [63]. Although we accessed the cell of origin of DLBCL in only 109 patients, we found 47.7% (52) of cases with a non-GCB phenotype in our cohort. These data are in agreement with what was previously reported in cohorts of patients from developed countries, despite being a population from another geographic area and with different ancestry.

Here, we also identified that age ≥ 75 years, advanced clinical stage III/IV, B-symptoms, high LDH levels, neutrophilia, and low Ly/Mo ratio were independent adverse prognostic factors associated with decreased OS and PFS in adults older than 70 years with DLBCL, NOS. These data are in agreement with those previously reported by other international collaborative groups. Characteristically, advanced age, elevated LDH, and Ann Arbor/Cotswold clinical stage III/IV are listed as variables associated with poor clinical outcomes in robust prognostic indices widely used in clinical practice for the risk stratification of individuals with DLBCL, such as the IPI, R-IPI, and NCCN-IPI [22,23,24]. The presence of constitutional symptoms, such as fever, weight loss, and night sweats (B-symptoms), indirectly reflects the lymphoma tumor burden, being associated with adverse outcomes in different subtypes of chronic lymphoproliferative disorders [64,65]. Some laboratory parameters, easily accessible to the peripheral blood count, have recently been incorporated into biological prognostic indexes and present a strong association with adverse clinical outcomes in cancer. Absolute neutrophil count has been considered a surrogate marker of inflammation, absolute lymphocyte count reflects host immunity, and absolute monocyte count, as well as the tumor-associated macrophage content, are surrogate markers of the tumor microenvironment. Therefore, some applications derived from these blood cells have been studied, including the neutrophil/lymphocyte ratio and lymphocyte/monocyte ratio, both associated with a poor prognosis in patients with DLBCL [66,67,68]. The identification of all these prognostic factors is of great importance in identifying older adults with DLBCL with a greater probability of therapeutic failure or early progression and who, consequently, may be candidates for intensified therapeutic strategies and associated with a greater chance of long-term survival, such as the full-dose R-CHOP regimen.

## 5. Conclusions

In this large and real-life South American cohort, we demonstrated that patients with DLBCL, NOS older than 70 years still do not have satisfactory clinical outcomes in 2024, with half of cases not reaching 5 years of life expectancy after diagnosis. Although the conventional R-CHOP offers response and survival advantages over attenuated regimens, such as R-MiniCHOP and R-MiniCHOP of the elderly, its myelotoxicity is not negligible. However, we highlight that a substantial portion of these individuals present great geriatric vulnerability, being ineligible for the full-dose R-CHOP regimen. This subgroup of patients should be promptly identified at the time of diagnosis and will be candidates for therapeutic strategies adapted to their frailty profile. Therefore, we hope that the outcomes reported, and the prognostic factors identified here, may assist clinicians in the appropriate selection of risk-adapted therapeutic strategies for old and very old DLBCL patients in a real-life setting, due to the fact that Comprehensive Geriatric Assessments are complex and unfeasible tools and are not routinely incorporated into clinical practice.

## Figures and Tables

**Figure 1 cancers-16-01459-f001:**
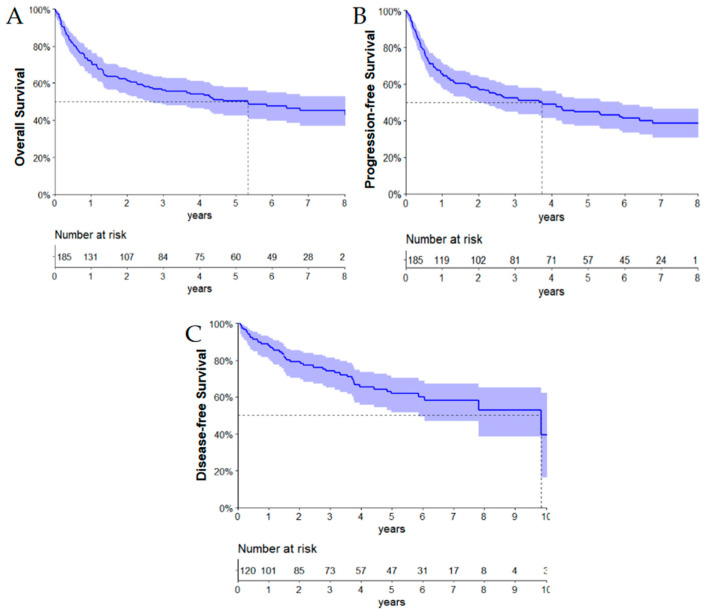
Survival curves for 185 Brazilian patients with DLBCL, NOS aged ≥ 70 years. (**A**) Overall survival, (**B**) progression-free survival, and (**C**) disease-free survival.

**Figure 2 cancers-16-01459-f002:**
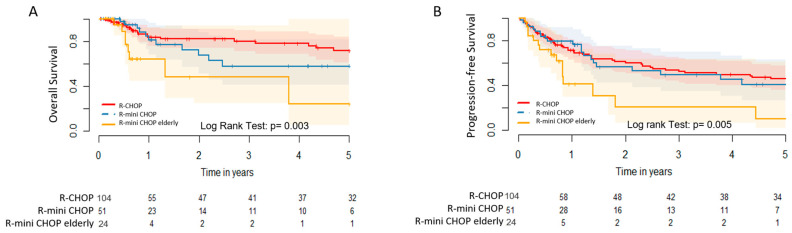
(**A**) OS curve (*p* = 0.003) and (**B**) PFS curve (*p* = 0.005) for 179 treated DLBCL, NOS patients older than 70 years according to up-front immunochemotherapeutic regimen. Red: conventional R-CHOP-21 regimen, blue: R-MiniCHOP-21 regimen, and yellow: R-MiniCHOP of the elderly regimen.

**Table 1 cancers-16-01459-t001:** Clinical–demographic and laboratorial characteristics of 185 Brazilian patients with DLBCL, NOS aged ≥ 70 years.

Characteristic	N = 185 (%; 95% CI)
Gender	
Male	76 (41.1%; 34.3–48.4%)
Female	109 (58.9%; 51.8–65.9%)
Median age (range)	75 years (70–97 years)
Caucasian ethnicity	142 (76.7%; 69.5–81.8%)
≥1 comorbidity	117 (63.2%; 56.2–70.1%)
Immobilism	36 (19.5%; 13.7–25.2%)
Polypharmacy (≥5 drugs)	46 (24.8%; 18.6–30.9%)
Desnutrition (geriatric BMI < 22)	52 (28.1%; 21.6–34.5%)
Categorization according to KPS	
Fit (KPS > 70)	104 (56.2%; 46.5–65.8%)
Unfit (KPS 50–70)	51 (27.6%; 15.3–39.8%)
Frail (KPS < 50)	30 (16.2%; 2.9–29.5%)
B-symptoms	117 (63.2%; 56.2–70.1%)
Bulky disease ≥ 7 cm	90 (48.6%; 41.4–55.7%)
Extranodal involvement	141 (76.2%; 70.0–82.3%)
≥2 extranodal sites	69 (37.2%; 30.2–44.1%)
BM infiltration	7 (3.8%; 1.0–6.5%)
CNS infiltration	7 (3.8%; 1.0–6.5%)
Advanced-stage III or IV	134 (72.4%; 66.0–78.7%)
Intermediate–high/high-risk IPI	124 (67.0%; 60.2–73.7%)
Intermediate–high/high-risk NCCN-IPI	163 (88.1%; 83.4–92.7%)
LDH ≥ 1.5 × UVN	48 (25.9%; 19.5–32.2%)
Hypoalbuminemia (<3.5 g/dL)	52 (28.6%; 22.0–35.1%)—N miss = 3
Hypogammaglobulinemia (<1.5 g/dL)	2 (1.1%; 0.4–1.7% )—N miss = 5
COO according to IHC (Hans algorithm) GCB-like phenotype ABC-like phenotype	N = 109 57 (52.3%; 42.9–61.9%) 52 (47.7%; 38.2–57.2%)

BMI: body mass index; KPS: Karnofsky Performance Status scale; BM: bone marrow; CNS: central nervous system; IPI: International Prognostic Index; NCCN-IPI: National Comprehensive Cancer Network International Prognostic Index; LDH: lactic dehydrogenase; UVN: upper value of normality; COO: cell of origin; IHC: immunohistochemistry; GCB: germinal center B-cell; ABC: activated B-cell; N miss: missing data.

**Table 2 cancers-16-01459-t002:** Up-front therapy modalities, characteristics, responses, and adverse event profile in 179 Brazilian patients with DLBCL, NOS aged ≥ 70 years.

	R-CHOP-21 (N = 104; 95% CI)	R-MiniCHOP (N = 51; 95% CI)	R-Mini-CHOP of the Elderly (N = 24; 95% CI)	*p*-Value
Median age	73 years	78 years	83 years	0.367 *
CCI score	5.5 points	6.0 points	6.0 points	0.998 *
LDH ratio > 1.5	21 (20.4%; 12.6–28.1%)	16 (31.4%; 18.6–44.1%)	7 (29.2%; 11.0–47.3%)	0.269 **
Albumin < 3.5g/dL	19 (18.8%; 11.2–26.3%)	16 (31.4%; 18.6–44.1%)	13 (54.2%; 34.2–74.2%)	0.001 **
IPI ≥ 3	60 (57.7%; 48.1–67.2%)	40 (78.4%; 67.0–89.7%)	19 (79.2%; 62.9–95.4%)	0.013 **
NCCN-IPI ≥ 3	83 (79.8%; 72.0–87.5%)	50 (98.0%; 94.1–100%)	24 (100%)	0.002 **
G3/4 neutropenia	103 (99.0%; 97.0–100%)	47 (92.2%; 84.8–99.5%)	20 (83.3%; 68.3–98.2%)	0.003 **
Febrile neutropenia	57 (54.8%; 45.2–64.3%)	27 (52.9%; 39.1–66.6%)	12 (50.0%; 29.9–70.1%)	0.907 **
G3/4 thrombocytopenia	22 (21.2%; 13.3–29.0%)	12 (23.5%; 11.8–35.1%)	9 (37.5%; 18.1–56.9%)	0.238 **
Therapy interruption	13 (12.5%; 6.1–18.8%)	4 (7.8%; 0.4–15.1%)	3 (12.5%; 6.2–18.8%)	0.671 **
Mortality during induction regimen	18 (17.3%; 10.1–24.5%)	10 (19.6%; 8.6–30.5%)	6 (25.0%; 7.7–42.2%)	0.681 **
ORR	75 (72.1%; 63.4–80.7%)	36 (70.6%; 58.1–83.0%)	11 (45.6%; 25.6–65.6%)	0.040 **

CCI: Charlson Comorbidity Index; LDH: lactic dehydrogenase; IPI: International Prognostic Index; NCCN-IPI: National Comprehensive Cancer Network International Prognostic Index; ORR: overall mortality rate; G3/G4: grade 3 or 4—according to the CTCAE version 5.0. * *p*-value was obtained by Kruskal–Wallis test; ** *p*-value was obtained by chi-square test.

**Table 3 cancers-16-01459-t003:** Main prognostic factors for OS and PFS identified in elderly DLBCL, NOS patients aged ≥ 70 years.

Variable	OS Univariate	OS Multivariate	PFS Univariate	PFS Multivariate
Age ≥ 75 years	HR: 2.08, 95% CI: 1.35–3.20, *p* = 0.001	HR: 2.22, 95% CI: 1.42–3.47, *p* = 0.001	HR: 2.00, 95% CI: 1.33–3.01, *p* = 0.001	HR: 2.33, 95% CI: 1.52–3.58, *p* < 0.001
Non-Caucasian ethnicity	HR: 1.80, 95% CI: 1.14–2.83, *p* = 0.012		HR: 1.68, 95% CI: 1.09–2.60, *p* = 0.020	
Neurologic comorbidity	HR: 2.37, 95% CI: 1.19–4.72, *p* = 0.014		HR: 2.76, 95% CI: 1.38–5.49, *p* = 0.004	
LDH > UVN	HR: 1.60, 95% CI: 1.07–2.41, *p* = 0.023		HR: 1.48, 95% CI: 1.00–2.17, *p* = 0.048	HR: 1.09, 95% CI: 1.00–1.18, *p* = 0.047
Leukocytosis	HR: 2.61, 95% CI: 1.59–4.28, *p* < 0.001		HR: 2.26, 95% CI: 1.38–3.69, *p* = 0.001	
Neutrophilia	HR: 2.32, 95% CI: 1.46–3.67, *p* < 0.001	HR: 2.18, 95% CI: 1.23–3.84, *p* = 0.007	HR: 2.09, 95% CI: 1.34–3.27, *p* = 0.001	HR: 2.25, 95% CI: 1.41–3.59, *p* = 0.001
Monocytosis	HR: 2.01, 95% CI: 1.17–3.02, *p* < 0.001		HR: 1.97, 95% CI: 1.24–2.73, *p* = 0.007	
L/M ratio < median	HR: 2.19, 95% CI: 1.32–3.64, *p* = 0.002	HR: 1.98, 95% CI: 1.18–3.32, *p* = 0.010	HR: 2.04, 95% CI: 1.25–3.33, *p* = 0.004	HR: 2.08, 95% CI: 1.25–3.46, *p* = 0.005
Hypoalbuminemia	HR: 1.59, 95% CI: 1.03–2.46, *p* = 0.036		HR: 1.50, 95% CI: 1.08–2.27, *p* = 0.049	
Ann Arbor CS III/IV	HR: 2.36, 95% CI: 1.38–4.04, *p* = 0.002	HR: 2.36, 95% CI: 1.34–4.13, *p* = 0.003	HR: 2.58, 95% CI: 1.53–4.34, *p* < 0.001	HR: 2.38, 95% CI: 1.38–4.11, *p* = 0.002
B-symptoms	HR: 1.89, 95% CI: 1.21–2.93, *p* = 0.005		HR: 1.84, 95% CI: 1.21–2.80, *p* = 0.004	HR: 1.62, 95% CI: 1.04–2.52, *p* = 0.032
≥2 extranodal sites	HR: 2.73, 95% CI: 1.19–6.26, *p* = 0.018		HR: 2.38, 95% CI: 1.04–5.45, *p* = 0.039	

OS: overall survival, PFS: progression-free survival, HR: hazard ratio; 95% CI: 95% confidence interval, *p*: *p*-value, LDH: lactic dehydrogenase, UVN: upper value of normality, L/M: lymphocyte/monocyte ratio, CS: clinical stage.

**Table 4 cancers-16-01459-t004:** Responses and clinical outcomes obtained in the main clinical trials involving newly diagnosed DLBCL older adults.

Modality	Trial	Phase	Treatment	N	ORR %	PFS %	OS %
Anthracycline full dose	LNH98-5 [32]	3	CHOP R-CHOP	399	69 83	30 (5y) 54 (5y)	45 (5y) 58 (5y)
	RICOVER [34]	3	R-CHOP + 2xR R-CHOP + 2xR + RT	166	76 (CR) 78 (CR)	72 (3y) 73 (3y)	77 (3y) 78 (3y)
	UK NCRI [37]	3	R-CHOP14 R-CHOP21	604	91 91	64 (5y)	69 (5y)
	LNH03-6B [35]	3	R-CHOP14 R-CHOP21	602	87 86	60 (3y) 62 (3y)	69 (3y) 72 (3y)
	Corazzelli et al., 2011 [38]	2	R-COMP14	41	73	77 (4y)	67 (4y)
Anthracycline dose attenuated	Peyrade et al., 2011 [29]	2	R-MiniCHOP of the elderly	150	73	47 (2y)	59 (2y)
	ANZINTER3 [39]	3	R-MiniCEOP* R-CHOP	224	81 87	46 (5y) 48 (5y)	63 (5y) 62 (5y)
Non-anthracycline	Park et al., 2016 [40]	2	BR	23	78	5.4 months (median)	10.2 months (median)
	Fields et al., 2014 [41]	2	R-GCVP	62	61.3	50 (2y)	56 (2y)
	Qui-Dan et al., 2018 [42]	2	R-GEMOX	60	49	49 (3y)	65 (3y)

CHOP (cyclophosphamide, doxorubicin, vincristine, and prednisone); R-CHOP (rituximab plus CHOP); R: rituximab; RT: radiotherapy; R-COMP (rituximab, cyclophosphamide, vincristine, liposomal doxorubicin, and prednisone); R-MiniCEOP* (epirubicin); mini-CEOP** (etoposide); BR (bendamustine plus rituximab); R-GCVP (rituximab, gemcitabine, cyclophosphamide, vincristine, and prednisone); R-GEMOX (rituximab, gemcitabine, and oxaliplatin); ORR: overall response rate; PFS: progression-free survival; OS: overall survival.

**Table 5 cancers-16-01459-t005:** Clinical outcomes obtained in selected studies involving older adults with DLBCL stratified according to geriatric vulnerability.

Study	Design	N	Median Age	Stratification	Outcomes
Tucci et al., 2015 Italy [15]	Prospective	173	77	Fit 46% Unfit 16% Frail 38%	2-year OS: Fit: 84% Non-fit 47%
Merli et al., 2014 Italy [44]	Prospective	318	Fit 72 Frail 78	Fit 70% Frail 30%	5-year OS: Fit: 71.5% Frail: 23.8% Whole cohort: 28%
Ong et al., 2019 Australia [45]	Retrospective	205	73	Fit 41% Unfit 21% Frail 38%	3-year OS: Fit: 82% Unfit: 60% Frail: 53%
Spina et al., 2012 Italy [46]	Prospective	100	75	Fit 55% Unfit 32% Frail 13%	5-year OS: Whole cohort: 60%
Olivieri et al., 2012 Italy [47]	Prospective	91	74	Fit 59% Unfit 24% Frail 17%	5-year OS: Whole cohort: 46% 5-year EFS: Whole cohort: 31%
Marchesi et al., 2013 Italy [48]	Retrospective	73	78	Fit 29% Unfit 38% Frail 33%	2-year OS: Fit: 58.3% Non-fit: 24.3% 2-year PFS: Fit: 47.2% Non-fit 21.6%
Lage et al., 2024 Brazil (This study)	Retrospective	185	75	* Fit 56.2% Unfit 27.6% Frail 16.2%	5-year OS: Fit: 71.6% Unfit: 57.8% Frail **: 24.1% Whole cohort: 50.2%

* Fitness according to KPS scale; ** Estimated 5-year OS considering just treated cases with anthracycline-containing regimens (three frail cases not treated and three frail cases treated with R-CVP were excluded); OS: overall survival; PFS: progression-free survival.

## Data Availability

All data generated and analyzed during this study were included in this published article. The raw data for this study are in possession of the correspondence author and may be fully available in the event of a request to the correspondence author via e-mail.

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
