# Peer review of "Predictors of Survival, Treatment Modalities, and Clinical Outcomes of Diffuse Large B-Cell Lymphoma in Patients Older Than 70 Years Still an Unmet Medical Need in 2024 Based on Real-World Evidence"

_cancers, 2024, doi:10.3390/cancers16081459_

Round 1

Reviewer 1 Report

Comments and Suggestions for Authors

The interesting manuscript entitled “Predictors of survival, treatment modalities, and clinical out-comes of diffuse large B-cell lymphoma in elderly patients older than 70 years – still an unmet medical need in 2024 based on real-world evidence” by de Pádua Covas Lage et al analyzed the clinical outcomes and identify predictors of survival among elderly patients diagnosed with diffuse large B-cell lymphoma (DLBCL). Specifically, the study aims to compare the effectiveness and adverse event profiles of different primary therapeutic modalities, focusing on immunochemotherapeutic regimens based on anthracycline agents at attenuated doses versus the conventional R-CHOP regimen. Through a thorough analysis of a large real-world cohort of elderly patients (185), the study aims to provide insights into the optimal upfront treatment strategies for this population, considering both treatment efficacy and patient tolerability.

The manuscript offers a comprehensive assessment introduction of clinical outcomes and predictors of survival in elderly patients with diffuse large B-cell lymphoma (DLBCL), focusing on comparing different primary therapeutic modalities. The introduction sets a clear framework for the study, highlighting the significance of the research question within the medical landscape.

The methodology section is well-structured and meticulously described (from the study design and ethics, eligibility criteria, staging procedures, current therapies, response assessment, histopathological diagnosis to the statistical analyses of the obtained data), allowing for transparency and reproducibility. The inclusion of a large, real-world cohort of elderly patients from Brazil spanning over a decade (2009-2020) enhances the generalizability of the findings. The comparative analysis between different therapeutic regimens, including R-CHOP and attenuated immunochemotherapy regimens, is appropriately conducted.

The results section presents a detailed analysis of clinical outcomes, response rates, and adverse events associated with different therapeutic modalities. The identification of independent prognostic factors adds depth to the analysis, enriching our understanding of DLBCL in elderly patients.

The discussion effectively synthesizes the findings, contextualizing them within existing literature and clinical practice. The comparison between full-dose R-CHOP and attenuated regimens sheds light on the trade-offs between treatment efficacy and tolerability in elderly patients. The conclusion aptly summarizes the key findings, emphasizing the unmet medical need for optimal upfront treatment strategies in this population.

Regarding the conclusions, the authors effectively highlight the ongoing challenges in achieving satisfactory outcomes for elderly DLBCL patients and underscores the need for personalized therapeutic strategies. Despite the superiority of conventional R-CHOP therapy, its myelotoxicity limits its applicability, particularly in frail individuals. The emphasis on prompt identification and risk-adapted treatment approaches based on real-life evidence addresses practical limitations in clinical practice. By identifying prognostic factors, the study offers valuable insights to guide clinicians in selecting appropriate interventions, ultimately aiming to improve outcomes in this vulnerable patient population.

Suggestions for improvement:

Expand on the potential implications of the study findings for the development of novel therapeutic approaches. This could involve exploring the feasibility of integrating biomarker-driven strategies or targeted therapies into clinical practice, with the aim of optimizing outcomes while minimizing treatment-related toxicity in this vulnerable population.

Discuss the implications of the findings on clinical practice guidelines and potential future directions for research, such as exploring novel treatment modalities tailored to the unique needs of elderly patients.

Author Response

Reviewer #1
Expand on the potential implications of the study findings for the development of novel therapeutic approaches. This could involve exploring the feasibility of integrating biomarkerdriven strategies or targeted therapies into clinical practice, with the aim of optimizing outcomes while minimizing treatment-related toxicity in this vulnerable population. Discuss the implications of the findings on clinical practice guidelines and potential future directions for research, such as exploring novel treatment modalities tailored to the unique needs of elderly patients.

Dear reviewer, as requested by you, in paragraphs 11 and 12 of the Discussion we add information regarding new therapeutic strategies for the management of elderly patients with DLBCL, as well as we make considerations about how the prognostic factors identified in our study could assist in the appropriate selection of elderly patients who are candidates for such therapies and for therapies adapted to their geriatric vulnerability. 

Reviewer 2 Report

Comments and Suggestions for Authors

This is a very interesting article because it is done in the real world and predicts the survival and results of diffuse large B-cell lymphoma treatments in elderly patients older than 70 year

Here you have several considerations that may improved  the mansucript

MAJOR ISSUES

1.       In my opinion to introduce in the text and in the table the absolute values of the numerator and denominator eg “63.2% (117/185)  “  It makes the text more difficult to read, without contributing anything.

2.       there are also some cases that have a negative value, and I don't quite understand what it refers to, for example "More than a quarter of the whole cohort (25.4% - 47/185) had diabetes".

3.       What does -47 mean? does it mean that there were 47 who did not have diabetes? In any case my suggestion is to eliminate the absolute values.

4.       It is preferable to calculate the 95% confidence intervals of the proportions.

5.       In table 1 eliminate the broken numbers (numerator and denominator) It is unnecessary because the denominator is most of the time the same 185 , furthermore is not very informative. It can be written in brackets in the column title under N.  In all the variables the quotient is presented first with the absolute values, and then the proportion. Except in the case of Advanced-stage III or IV where the order is reversed. 72.4% (134/185) Is this a typo or does it mean something? Instead of these n  numbers, the 95% confidence interval of the proportions should be put in parentheses. This parameter considers the sample size and is more informative than the n. You can calculate it with the free program openepi, (https://www.openepi.com/Proportion/Proportion.htm )  or with any other software

6.       In table 2, the authors present comparisons between 3 groups ( 8R-CHOP-21,0R-MiniCHOP, R-Mini-CHOP of the elderly) and there is a P. But no where neither in in the results, nor in the table we can know what statistical test was done. From the material and methods it can be deducted that is a chi square. The authros should write a note at the foot of the table

7.       In table 3, Median age and CCI score has no p-value. If these are quantitative variables, a chi square can not be performed, but a nonparametric test such as Kruskall Wallis should be performed.

8.       In my opinion, the probabilities should be eliminated in Table 3; it is enough to provide the HR and the 95% confidence interval.

 MINOR ISSUES

9.       According to the instructions for authors “: References must be numbered in order of appearance in the text (including table captions and figure legends) and listed individually at the end of the manuscript” The authors are incuding within the text the in brackets the reference’s first authors, you should use a numbered citation instead.

Author Response

  1. In my opinion to introduce in the text and in the table the absolute values of the numerator and denominator e.g. “63.2% (117/185) “ It makes the text more difficult to read, without contributing anything.

As suggested by the reviewer, the denominator of the absolute values presented in the results section were removed from the text and tables, except for the variables where there was missing data, and therefore the denominator would be different from 185.
2. There are also some cases that have a negative value, and I don't quite understand what it refers to, for example "More than a quarter of the whole cohort (25.4% - 47/185) had diabetes".

Dear reviewer, these cases do not refer to negative values, but rather a dash (-) was used to separate the percentage from the absolute values. To avoid such confusion, we chose to replace the dash by a semicolon (;) e.g. 25.4%; 47 (%; n)

3. What does -47 mean? does it mean that there were 47 who did not have diabetes? In any case my suggestion is to eliminate the absolute values.

As explained in the previous question, these are not negative values, and as suggested by the reviewer, the denominator of absolute values was removed in order to facilitate reading of the text.

4. It is preferable to calculate the 95% confidence intervals of the proportions.

The 95% confidence interval was calculated for all proportions and added to Tables 1 and 2.

5. In table 1 eliminate the broken numbers (numerator and denominator) It is unnecessary because the denominator is most of the time the same 185, furthermore is not very informative. It can be written in brackets in the column title under N. In all the variables the quotient is presented first with the absolute values, and then the proportion. Except in the case of Advanced-stage III or IV where the order is reversed. 72.4% (134/185) Is this a typo or does it mean something? Instead of these n numbers, the 95% confidence interval of the proportions should be put in parentheses. This parameter considers the
sample size and is more informative than the n. You can calculate it with the free program openepi, (https://www.openepi.com/Proportion/Proportion.htm) or with any other software

As requested by the reviewer, in Table 1 the denominator of absolute values was removed. The N (N=185) was added in square brackets in the title column of the same table. In relation to the Advanced-stage III or IV, the order of relative and absolute values was inverted, in order to be standardized in relation to that presented for all other variables. Finally, the 95% CI for all proportions was calculated and added in parentheses.

6. In table 2, the authors present comparisons between 3 groups (R-CHOP-21, R-MiniCHOP, R-Mini-CHOP of the elderly) and there is a p. But nowhere neither in the results, nor in the table we can know what statistical test was done. From the material and methods, it can be deducted that is a chi square. The authors should write a note at the foot of the table.

In the Statistical Analysis section and at the foot of the Table 2, information was added regarding the statistical tests used (chi-square test and Kruskal-Wallis test) to assess whether there were statistical differences between clinical-demographic variables and responses
compared in three main treatment groups

7. In table 2, Median age and CCI score has no p-value. If these are quantitative variables, a chi square cannot be performed, but a nonparametric test such as Kruskall Wallis should be performed.

In Table 2, the median age and CCI score had their p-values calculated using the KruskalWallis test, as suggested by the reviewer. These values were plotted in the aforementioned table.

8. In my opinion, the probabilities should be eliminated in Table 3; it is enough to provide the HR and the 95% confidence interval.

Dear reviewer, table 3 does not contain probabilities, it only contains hazard ratio (HR), 95% confidence interval (95% CI) and p-value.

9. According to the instructions for authors “References must be numbered in order of appearance in the text (including table captions and figure legends) and listed individually at the end of the manuscript” The authors are including within the text the in brackets the reference’s first authors, you should use a numbered citation instead.

In accordance with the standards of the scientific journal Cancers, references were numbered in the text according to the order of appearance and listed at the end of the manuscript. 

Reviewer 3 Report

Comments and Suggestions for Authors

The original article “Predictors of survival, treatment modalities, and clinical outcomes of diffuse large B-cell lymphoma in elderly patients older than 70 years – still an unmet medical need in 2024 based on real-world evidence” reported that the difficulty of R-CHOP treatment for very elderly DLBCL using Brazilian real-world data. This article was reflected daily practice and showed that the optimal schedule of R-CHOP regimen for elderly DLBCL remained unclear. I considered that this article could be suitable for acceptance for “Cancers”, but there were several issues before that.

1.       The volume of introduction and discussion parts were quite heavy. The author should reduce the volume of them.

2.       The early phase mortality due to infection was high in your data. Was prophylaxis for infection, such as antibiotics and G-CSF, unified? If not, was there the difference of prophylaxis among these three groups?

3.       The relative dose intensity of R-CHOP was associated with clinical outcome according to the previous reports. Do the authors have any data about RDI? If yes, did the RDI affect ORR, PFS and OS in your dataset?

4.       There was significant difference of ORR and survival time among these three groups. When the treatment response was similar, were the PFS and OS similar among these three groups? (For instance, when CR was achieved, the PFS and OS were similar among R-CHOP, R-miniCHOP, and R-miniCHOP for very elderly patients?

Author Response

  1. The volume of introduction and discussion parts were quite heavy. The author should reduce the volume of them.

Dear reviewer, although the volume of the Introduction and Discussion sections are indeed very dense, this is justified due to the high number of studies and information related to this topic available in current medical literature. With the information available in the Introduction, we seek to provide a broad overview of the challenges of treating DLBCL in elderly and very elderly patients, as well as the main current therapeutic possibilities for this special population. In the Discussion, we address the different clinical and real-life studies involving primary treatment of elderly patients with DLBCL, compare the outcomes of our
cohort with those of other international cohorts and outline an overview of therapeutic strategies adapted to risk regarding the independent prognostic factors found in our study, with a focus on assisting personalized therapeutic decisions for geriatric patients with DLBCL presenting different frailty profiles. Therefore, due to the broad aspects covered in these sections, we justify the dense load of information contained in these parts of the manuscript.

2. The early phase mortality due to infection was high in your data. Was prophylaxis for infection, such as antibiotics and G-CSF, unified? If not, was there the difference of prophylaxis among these three groups?

Dear reviewer, indeed early mortality rates were high in our study, however our
population consisted of elderly and very elderly DLBCL patients with a high geriatric vulnerability profile, as presented in our Results (section 3.1. - Clinical, Epidemiological, Laboratory and Histopathological features). Prophylaxis for infections was unified across the three main different treatment regimens
and included use of granulocyte growth factors (G-CSF) for individuals with absolute neutropenia < 1.0 x 109 /L, as well as universal prophylaxis for pneumocystosis with trimethoprimsulfamethoxazole 400/80 mg P.O. per day and secondary prophylaxis for herpes zoster with acyclovir 400 mg P.O. twice per day. Antibacterial prophylaxis with quinolones was not routinely adopted for either treatment group. This last paragraph was added to the body of the text.

3. The relative dose intensity of R-CHOP was associated with clinical outcome according to the previous reports. Do the authors have any data about RDI? If yes, did the RDI affect ORR, PFS and OS in your dataset?

Dear reviewer, although this is an interesting and pertinent question, unfortunately we do not have data regarding relative dose intensity (RDI) for the R-CHOP regimen in this study. As this was a retrospective study, where the evaluation and calculation of RDI was not within the scope of our objectives, this data was not accessed. However, as reported in the manuscript, we have information regarding dose delays and therapy interruption. Concerning to these variables, these rates were not statistically different between the three main immunochemotherapeutic regimens (see Table 2).

4. There was significant difference of ORR and survival time among these three groups. When the treatment response was similar, were the PFS and OS similar among these three groups? (For instance, when CR was achieved, the PFS and OS were similar among R-CHOP, R-miniCHOP, and R-miniCHOP for very elderly patients?

Dear reviewer, as suggested by you, we conducted an additional survival analysis in an exploratory way involving only cases with a complete response (CR) at the end of primary therapy. Considering DLBCL cases that achieved CR after first-line therapy, the 2-year and 5-year PFS estimates were 87.4% (95% CI: 79.0-96.6%) and 66.7% (95% CI: 55.2-80.6%) for R-CHOP, and 75.0% (95% CI: 58.2-96.6%) and 56.3% (95% CI: 37.1-85.3%) for R-MiniCHOP, p=0.200. Similarly, for this
same population, the 2-year and 5-year OS estimates were 92.8% (95% CI: 86.3-99.9%) and 81.0% (95% CI: 70.4-93.2%) for R-CHOP, and 79.7% (95% CI: 63.5-99.9%) and 68.3% (95% CI: 50.0-93.2%) for R-MiniCHOP, p=0.400. In this analysis, the group treated with R-miniCHOP of the elderly could not be included in the comparison, because the number of patients treated with this regimen who
achieved CR was very low (n=8/24), not allowing a reliable comparison. Therefore, the important imbalance between the three therapeutic groups and the extremely restricted sample size for the R-MiniCHOP of the elderly arm made an accurate and reliable analysis unfeasible. 

Reviewer 4 Report

Comments and Suggestions for Authors

This article is interesting because it highlights the challenges of chemotherapy in older patients by comparing the effectiveness of reduced anthracycline regimens with conventional regimens in patients with DLBCL aged 70 years and older.

1. This study suggests that the poor treatment outcomes of R-MiniCHOP and R-MiniCHOP of the elderly may be due to basic information such as the nutritional status and age of the patients for whom such reduced regimens were selected. The authors should be more specific about how such information should be used in future treatment strategies.

2. Did peripheral neuropathy caused by vincristine affect the QOL of elderly patients?

3. Consideration is needed as to whether BiTE and CAR-T cell therapy can be alternative treatments for DLBCL in the elderly.

Author Response

  1. This study suggests that the poor treatment outcomes of R-MiniCHOP and R-MiniCHOP of the elderly may be due to basic information such as the nutritional status and age of the patients for whom such reduced regimens were selected. The authors should be more specific about how such information should be used in future treatment strategies.

Dear reviewer, as requested by you, in paragraphs 11 and 12 of the Discussion we add information regarding new therapeutic strategies for the management of elderly patients with DLBCL, as well as we make considerations about how the prognostic factors identified in our study could assist in the appropriate selection of elderly patients who are candidates for such therapies.

2. Did peripheral neuropathy caused by vincristine affect the QOL of elderly patients?

Although peripheral neuropathy associated with vincristine use is a limiting factor for application of R-CHOP-like regimens, particularly in elderly individuals, this issue was not evaluated in our study. This is basically due to the fact that it is a retrospective study, where the application of quality of life questionnaires (QOL) was not possible.

3. Consideration is needed as to whether BiTE and CAR-T cell therapy can be alternative treatments for DLBCL in the elderly.

Yes, CAR-T therapies and bispecific monoclonal antibodies constitute important
therapeutic alternatives for a select subgroup of elderly patients with DLBCL. As requested by the reviewer, two paragraphs (paragraphs 11 and 12) were added to the Discussion section of the manuscript addressing this topic.

Finally, we would like to thank you once again. We look forward to hearing from you regarding our submission. We would be glad to respond to any further questions and comments that you may have. 

Round 2

Reviewer 2 Report

Comments and Suggestions for Authors

The authors answered all the questions that were asked in the review.

Author Response

Dear Reviewer,

We want to thank you for your reviewers and contributions.
They were extremely important to improve both this work and future work with real-world data and evidence in patients with lymphoma.

Thank you very much.

Reviewer 3 Report

Comments and Suggestions for Authors

I considered that this revised article was described well, and so can be suitable for acceptance for "Cancers".

Author Response

Dear Reviewer,

We want to thank you for your reviewers and contributions.
They were extremely important to improve both this work and future work with real-world data and evidence in patients with lymphoma.
Attached is the final version of the manuscript.
Thank you very much,
